# Evaluation of Post-Hospital Care of Traumatic Brain Injury in Children, Adolescents and Young Adults—A Survey among General Practitioners and Pediatricians in Germany

**DOI:** 10.3390/diagnostics12092265

**Published:** 2022-09-19

**Authors:** Wiebke Käckenmester, Claas Güthoff, Dana Mroß, Gertrud Wietholt, Kristina Zappel, Ingo Schmehl

**Affiliations:** 1Center for Clinical Research, BG Klinikum Unfallkrankenhaus Berlin gGmbH, Warener Straße 7, 12683 Berlin, Germany; 2Kinderneurologie-Hilfe Berlin, Brandenburg am Unfallkrankenhaus Berlin, Warener Straße 7, 12683 Berlin, Germany; 3Bundesverband Kinderneurologie-Hilfe e.V., Tannenbergstraße 1, 48147 Münster, Germany; 4Department of Neurology, BG Klinikum Unfallkrankenhaus Berlin gGmbH, Warener Straße 7, 12683 Berlin, Germany

**Keywords:** traumatic brain injury, long-term effects, pediatrics, physician survey, post-hospital care, evaluation

## Abstract

Background: The long-term effects of mild Traumatic Brain Injury (TBI) in children and adolescents are increasingly discussed due to their potential impact on psycho-social development and education. This study aims to evaluate post-hospital care of children and adolescents after mild TBI using a physician survey. Methods: A self-developed, pre-tested questionnaire on diagnostics and treatment of TBI in outpatient care was sent to a representative sample of general practitioners and pediatricians in Germany. Results: Datasets from 699 general practitioners, 334 pediatricians and 24 neuropediatricians were available and included in the analysis. Nearly half of the general practitioners and most pediatricians say they treat at least one acute pediatric TBI per year. However, a substantive proportion of general practitioners are not familiar with scales assessing TBI severity and have difficulties assessing the symptoms correctly. Pediatricians seem to have better knowledge than general practitioners when it comes to treatment and outpatient care of TBI. Conclusions: To increase knowledge about TBI in outpatient physicians, targeted training courses should be offered, especially for general practitioners. Moreover, handing out written information about long-term effects and reintegration after TBI should be encouraged in outpatient practice.

## 1. Introduction

Traumatic Brain Injury (TBI) is one of the main causes for hospital admissions and deaths caused by accidents in children and adolescents in Germany. While the overall incidence is estimated at 332/100.000, TBIs are assumed to be more frequent in minors with 581/100.000 [1]. Nonetheless, 95% of TBIs in hospitalized patients are classified as mild [2,3,4].

Delayed or long-term effects resulting from mild TBI are discussed increasingly [4,5]. Dependent on diagnostics and population, it was estimated that 11 to 82% of patients showed symptoms that persisted longer than one month after trauma [6]. Sleep disturbances and fatigue [7], anxiety disorders and depression [8], ADHD [8,9] and cognitive impairment [9] are the most frequently described late effects of mild and moderate TBI. Further possible late effects are posttraumatic epilepsy [10], neuro-endocrinological disorders [11], posttraumatic stress disorders [8,9,12], impaired health-related quality of life [13], as well as implications on academic performance [9,14]. Due to mostly subtle initial symptoms, mild TBIs bear the risk that long-term effects remain undetected [8,9]. A large cohort study indicates that mild pediatric TBIs are associated with an increased probability of psychological disorders, lower educational level, and occupational disability in adulthood [14,15].

After hospital discharge with persisting symptoms after mild TBI, most German parents consult pediatricians or general practitioners [15]. Especially in rural areas, medical care of children and adolescents often takes place in general practice so that general practitioners should be prepared for acute pediatric TBI and possible long-term effects [16]. Therefore, outpatient care by pediatricians and general practitioners is of special importance for early detection of possible long-term effects after mild TBI. Post-hospital care and management of long-term effects in Germany have not been subjects of evaluation yet. The same applies to adherence to international guidelines with regard to reintegration and rehabilitation after TBI.

This study aims to evaluate outpatient care of children, adolescents and young adults who suffered a mild TBI by conducting a nationwide postal survey targeting self-reported expertise as well as knowledge of recent recommendations. Further topics of investigation are differences between general practitioners, pediatricians and neuropediatricians in self-reported expertise and knowledge as well as the association of self-reported expertise and knowledge.

## 2. Materials and Methods

In order to obtain a representative dataset, random samples of general practitioners and pediatricians were drawn from all registered physicians in Germany. In addition, all neuropediatricians in a managing role in sociopediatric centers were selected as they function as specialized therapists for TBI aftercare.

Questionnaires were sent by mail with the request to send back the completed form using the enclosed envelope. Four weeks after dispatch, 10% of the initially contacted physicians received a reminder via email. Data collection took place from May to December 2021 and was conducted anonymously. The study was reviewed by the regional ethics committee and registered in the German register of clinical studies (Deutsches Register Klinischer Studien DRKS).

### 2.1. Study Design

The questionnaire was sent out to 22% of the population of general practitioners and pediatricians, respectively, which resulted in 7699 contacted general practitioners and 1549 contacted pediatricians. The random sample was drawn clustered by federal states to ensure representativeness across the whole territory of Germany. Additionally, 154 neuropediatricians in managing roles in sociopediatric centers were all contacted. Selection of participants was conducted via online accessible databases of the Association of Statutory Health Insurance Physicians (Kassenärztliche Vereinigungen).

### 2.2. Questionnaire

The questionnaire consists of 54 items and was tested and optimized with a sample of the target population in a pilot study. The final version includes 54 items (Appendix A). The first five items represent the sample description (age, gender, medical specialty, duration and area of occupation) followed by items regarding the number of TBI patients per year, ways of referral, professional networks, attendance of trainings, knowledge of scales for assessing the severity of TBI and knowledge of guidelines. Furthermore, self-reported frequency of various diagnostic and awareness-raising measures were assessed. Symptoms of acute TBI and long-term effects have to be categorized as typical vs. untypical. Furthermore, 11 statements regarding pathogenesis, diagnostics, and aftercare of TBI had to be assessed for correctness.

### 2.3. Statistical Analysis

Questionnaires that were missing a specification of medical specialty were excluded because all analyses were conducted separated by specialty. Items surveying knowledge were summed up by counting correct answers. All answers to self-reported competence were assigned values from one to four with higher values indicating greater agreement. Only fully completed sets of items were included in sum score analyses.

Descriptive data were analyzed using mean and standard deviation in case of normal data and relative frequencies for categorical data. Subgroup analysis was carried out using Kruskal–Wallis-tests with Bonferroni adjusted Dunn tests for multiple post hoc comparisons. Correlations were assessed by Spearman’s Rho. The significance level for all tests was set at 5%.

## 3. Results

The overall response rate was 12% (general practitioners 9%, pediatricians 22% and neuropediatricians 16%) with 1139 returned questionnaires. After exclusion of questionnaires according to the above-stated exclusion criteria, a final database of 1057 cases was set up with *n* = 699 general practitioners, *n* = 334 pediatricians and *n* = 24 neuropediatricians.

### 3.1. Sample Description

About half of the respondents were female (51%) and most were 45–65 years old (67%). The majority stated that they had worked for at least 15 years in their respective specialty (62%). A total of 20% of the respondents were established in a rural region, 22% in a small town, 27% in a middle-size town and 28% in a city. However, more general practitioners worked in rural regions (28%) in contrast to only 5% of pediatricians and neuropediatricians. Gender, age and duration of occupation were evenly distributed among specialties.

### 3.2. Self-Assessment

#### 3.2.1. Number of TBI Patients

During the last 12 months, approximately half of the general practitioners (53%) saw at least one child or adolescent with acute TBI in their practice and 52% stated the same for after-care treatment. Almost all pediatricians (93%) saw at least one acute pediatric TBI and 76% said they provided after care in the past 12 months. Neuropediatricians treated acute pediatric TBI in the last 12 months in 63% of cases and after care in 100%. Patient numbers are listed in detail in Table 1.

#### 3.2.2. Referral and First Presentation

Patients were most frequently referred from surgeons to general practitioners as well as pediatricians (see Table 2) while neuropediatricians stated referrals from pediatricians most frequently. However, many patients presented themselves without referral which was mostly initiated by their parents.

#### 3.2.3. Networks

While general practitioners stated neurologists, pediatricians and children’s hospitals as most prominent in their professional networks, pediatricians referred to neuropediatricians, child and adolescent psychiatrists and other pediatricians (see Table 3).

#### 3.2.4. Postgraduate Training 

26% of general practitioners, 51% of pediatricians and 88% of neuropediatricians self-reportedly took at least one postgraduate training on TBI.

#### 3.2.5. Knowledge of Severity Assessment Scales and TBI Guidelines

About half of general practitioners (54%) agreed to be aware of severity assessment scales and 30% quoted a correct scale. A total of 77% of pediatricians agreed with 50% giving accurate statements and all neuropediatricians agreed with 63% stating a correct scale. All respondents referred almost exclusively to the Glasgow Coma Scale. 

Knowledge of guidelines regarding pediatric TBI was stated by 80% of general practitioners with 4% quoting a correct guideline, by 56% of pediatricians with 25% with accurate statements and 86% of neuropediatricians with 54% stating a correct guideline. All respondents referred almost exclusively to the guideline of the AWMF.

#### 3.2.6. Recommendations and Interventions

The majority of general practitioners (81%) stated to give recommendations for the first 48 hours after an acute TBI. Most recommendations referred to observing symptoms as well as physical and mental rest. About the same recommendations were quoted by 80% of pediatricians as well as 63% of neuropediatricians with 94%, respectively, 75% stating to give recommendations. 

Only 3% of general practitioners, 12% of pediatricians and 42% of neuropediatricians stated they give recommendations for returning to school or job after TBI. 

Giving information about symptoms that require reexamination and giving information about long-term effects were mostly stated as always performed by respondents of all specialties. See Table 4 for a complete listing of frequency of interventions.

### 3.3. Knowledge and Self-Assessment

Nine symptoms had to be assessed for if they were typical for an acute TBI, with amnesia, sedation, vomiting, imbalance, headache, sensitivity to light and vertigo as typical symptoms and heightened appetite and urge to speak as untypical symptoms. For long-term effects, seven symptoms, with memory disorders, concentration disorders, nervousness/irritability, sleep disorders and behavioral disorders as typical symptoms and back pain as an atypical symptom, had to be judged. In a third block of knowledge items, the correctness of eleven statements regarding TBI had to be judged. Self-assessment regarding TBI management was surveyed with six four-level Likert items.

Subgroup analyses of acute TBI knowledge items revealed a significant difference between groups (H = 34.4, *p* < 0.001). Post hoc comparisons showed a significant difference only between general practitioners and pediatricians (*p* < 0.001) with the latter classifying more symptoms correctly. Analyses of long-term TBI effects knowledge items also revealed a significant difference between groups (H = 8.6, *p* = 0.014) with a significant difference between general practitioners and neuropediatricians (*p* = 0.023). Neuropediatricians classified more long-term effects correctly. No significant differences between groups were found regarding the third block of knowledge items. Self-assessment differed significantly between groups (H = 108.7, *p* < 0.001). Neuropediatricians assessed their competencies higher than general practitioners (*p* < 0.001) and pediatricians (*p* = 0.024) and pediatricians higher than general practitioners (*p* < 0.001).

## 4. Discussion

This study is the first data collection in Germany systematically evaluating the outpatient aftercare of TBI from a practitioner perspective using a postal survey and a representative sample of established general practitioners, pediatricians and neuropediatricians.

Nearly half of the general practitioners and most pediatricians of the survey sample treated at least one patient with an acute pediatric TBI while long-term effects seemed to be treated slightly less often. Given the incidence of 581/100.000 TBI among children and adolescents, this does not surprise and stresses the need for general practitioners and pediatricians to be prepared for outpatient TBI care. However, the survey showed that many general practitioners are not aware of the Glasgow Coma Scale and do not assess all typical symptoms correctly. However, most respondents of both specialties know and give correct recommendations for the first few days after an acute TBI and also provide information about symptoms requiring follow-up visits. Recommendations for returning to school or a job seem to be given much less, as well as handing out written information and contact data of specialized physicians. 

Pediatricians with and without neuropediatric specialization appeared to be better informed about TBI treatment than general practitioners. Nearly all pediatricians stated to have treated at least one acute TBI in the last 12 months. In contrast, only half of the general practitioners did. While this also could reflect a tendency of pediatricians to diagnose TBI more easily than general practitioners, this could, besides the medical specialization, reflect differences in knowledge about diagnostics and treatment. Differences regarding professional networks and attendance of training could also be due to the fact of general practitioners being less frequently established in urban regions. 

The results of the survey should be useful to optimize outpatient care of TBI for children, adolescents and young adults and facilitate the conceptualization of postgraduate training and information material. Increasing training opportunities seems to have particular potential as 75% of general practitioners and half of pediatricians stated to have not attended a TBI-specific training yet. Further expansion of professional networks involving practitioners and information centers could also have a beneficial effect on improvement in outpatient care.

Although recommendations for reintegration to school, kindergarten or jobs already exist, many professionals do not seem familiar with them. Written recommendations about reintegration, further treatment options and possible long-term complications could be useful to inform practitioners as well as patients and parents.

A limiting factor for the study was the low response rate of 9% among general practitioners which is unfortunately common in physician surveys. Moreover, physician surveys only may reflect a specific aspect of care conditions and come along with other biases as opposed to surveys focusing on the perspective of patients or parents. In particular, evaluation of care quality by patients seems to reflect attributes of the physician and of physician–patient communication [15] and, therefore, neglects other aspects of professional quality. Nonetheless, the perspective of patients and parents as well as objective data such as health insurance company data might be useful complements to this survey.

## Figures and Tables

**Table 1 diagnostics-12-02265-t001:** Numbers of Patients with acute TBI per year among the different specialties.

Acute TBI	Gen. Practitioner	Pediatrician	Neuro- Pediatrician	After Care	Gen. Practitioner	Pediatrician	Neuro- Pediatrician
/Year				/Year			
0	46%	5%	29%	0	53%	22%	0%
1–5	41%	17%	21%	1–5	38%	39%	38%
6–10	7%	13%	8%	6–10	4%	11%	25%
11–15	3%	11%	13%	11–15	3%	8%	13%
16–20	2%	18%	13%	16–20	2%	7%	13%
21–50	0.3%	22%	8%	21–50	1%	8%	13%
>50	0%	11%	4%	>50	0%	4%	0%

**Table 2 diagnostics-12-02265-t002:** Distribution of referrals and presenters of patients with acute TBI among the different specialties.

Most Frequent	Gen. Practitioner	Pediatrician	Neuro- Pediatrician	Most Frequent	Gen. Practitioner	Pediatrician	Neuro- Pediatrician
Referral				Presenter			
Surgeon	14%	15%	8%	Parents	41%	79%	54%
Pediatrician	2%	10%	71%	Teacher	1%	2%	0%
Rehabilitation Clinic	3%	2%	4%	Self Referral	1%	3%	0%
Gen. Practitioner	3%	4%	0%	Social Institution	0.3%	0.3%	0%
Neurologist	1%	0.3%	0%	Sports Coach	0.1%	0%	0%
other	14%	33%	4%	other	3%	7%	42%
n.s.	62%	36%	13%	n.s.	46%	9%	4%

**Table 3 diagnostics-12-02265-t003:** Network contacts as stated by the different specialties.

Network Contacts	Gen. Practitioner	Pediatrician	Neuro- Pediatrician
Neurologists	68%	24%	33%
Neuropsychologists	5%	2%	50%
Neuropediatricians	10%	83%	79%
Surgeons	56%	53%	46%
Pediatricians	56%	57%	79%
Child and Adolescent Psychiatrists	32%	66%	83%
Child and Adolescent Psychotherapists	21%	55%	75%
Youth Welfare Offices	24%	56%	58%
Kinderneurologie-Hilfe (Neurological Assistance for Children)	3%	6%	13%
Social-Pediatric Center	22%	79%	79%
Children’s Hospitals	53%	87%	92%
Other	5%	4%	13%

**Table 4 diagnostics-12-02265-t004:** Frequencies of interventions among the different specialties.

Frequency of Interventions		Gen. Practitioner	Pediatrician	Neuro- pediatrician
Information about symptoms requiring reexamination	Never	31.3%	2.4%	14.3%
	Rarely	11.2%	2.7%	14.3%
	Often	5.4%	8.5%	14.3%
	Always	52.1%	86.4%	57.1%
Handout of written information material	Never	87.0%	46.6%	52.4%
	Rarely	8.6%	20.7%	38.1%
	Often	2.4%	13.1%	4.8%
	Always	1.9%	19.5%	4.8%
Handout of specialist contacts	Never	39.4%	16.1%	23.8%
	Rarely	18.7%	38.3%	42.9%
	Often	26.5%	32.8%	23.8%
	Always	15.4%	12.8%	9.5%
Follow-up appointment	Never	34.6%	12.8%	14.3%
	Rarely	18.7%	54.0%	33.3%
	Often	26.8%	25.3%	33.3%
	Always	19.8%	7.9%	19.0%
Information about long-term effects	Never	31.8%	6.4%	14.3%
	Rarely	12.4%	13.9%	14.3%
	Often	13.4%	21.2%	28.6%
	Always	42.4%	58.5%	42.9%
Advice to inform school	Never	56.8%	27.1%	25.0%
	Rarely	21.3%	43.2%	30.0%
	Often	12.7%	20.1%	35.0%
	Always	9.2%	9.7%	10.0%

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
