# Peer review of "Evaluation of Post-Hospital Care of Traumatic Brain Injury in Children, Adolescents and Young Adults—A Survey among General Practitioners and Pediatricians in Germany"

_diagnostics, 2022, doi:10.3390/diagnostics12092265_

Round 1

Reviewer 1 Report

Review of manuscript titled ”Evaluation of post-hospital care of traumatic brain injury in children, adolescents and young adults – a survey among general practitioners and pediatricians in Germany”. The authors have in their study evaluated expertise and recommendations in children, adolescents and young adults who experienced a traumatic brain injury (TBI). Their study is relevant because it’s addressing the need for improved knowledge and management of TBI, primarily for general practitioners but also neuropaediatricians and paediatricians.

Abstract

How many general practitioners and paediatricians were included/did respond?

Any rehabilitations centres included?

Methods

Which were the exclusion criteria?

Author Response

Dear colleague,

thank you very much for your review.

  • The number of datasets included in the analysis has been inserted in the abstract.
  • As outpatient care after TBI was the focus of our investigation, no rehabilitation clinics were addressed in the study.
  • No formal inclusion or exclusion criteria regarding the respondents were defined in this study.

Best regards.

Reviewer 2 Report

Thank you for the opportunity to review this article. The study, by means of questionnaires for general practitioners, pediatricians and neuropediatricians, aims to provide a cross-sectional picture of knowledge regarding post-hospital management of traumatic brain injury in children and its sequelae.

The study addresses an interesting topic with little evidence in the literature, so I think it can provide a valid contribution to the field.

Methodologically, the study is well conducted and the article well written. In particular, the introduction is comprehensive, the methods and statistics well described, the results clear, the discussion detailed, and the conclusions consistent with the results.

Therefore, I think the article is worthy of publication in its current form. 

A little note: the acronym TBI in the first sentence was not introduced.

Thank you.

Author Response

Dear colleague,

thank you very much for your review.

The introduction of the acronym TBI has been inserted.

Best regards.